# Episodic future thinking in type 2 diabetes: Further development and validation of the Health Information Thinking control for clinical trials

Jeremiah M. Brown[1,2], Warren K. Bickel[1], Leonard H. Epstein[3], Jeffrey S. Stein[1,2]*

**1** Fralin Biomedical Research Institute at VTC, Roanoke, Virginia, United States of America, **2** Department of Human Nutrition, Foods, and Exercise, Virginia Tech, Blacksburg, Virginia, United States of America, **3** University at Buffalo Jacobs School of Medicine and Biomedical Sciences, Buffalo, New York, United States of America

* jstein1@vtc.vt.edu

## Abstract

Episodic Future Thinking (EFT) reduces delay discounting and may have the potential as a clinical tool to increase the likelihood of health-promoting behaviors. However, evaluations of EFT in clinical settings require control conditions that match the effort and frequency of cue generation, as well as participants' expectations of improvement. The Health Information Thinking (HIT) control addresses these issues, but how this control affects delay discounting in individuals with diabetes and obesity when utilizing diabetes-management specific health-information vignettes is unknown. Moreover, little research has explored whether EFT reduces delay discounting in individuals with type 2 diabetes. To this end, we examined the impact of EFT, HIT, and a secondary no-cue control condition (NCC; assessments as usual) on delay discounting in 434 adults with self-reported type 2 diabetes and obesity recruited using Amazon Mechanical Turk. After completing an initial screening questionnaire, eligible participants reported demographics, then were randomized to EFT, HIT, or NCC conditions. Following the generation of seven EFT or HIT cues, participants assigned to EFT or HIT conditions completed a delay discounting task while imagining EFT or HIT cues; no-cue participants completed the task without cues. EFT participants demonstrated significantly lower delay discounting levels than HIT or NCC participants; no differences in delay discounting between HIT and NCC participants were observed. These results suggest that engaging in EFT, but not diabetes-specific HIT, results in lower delay discounting in adults with type 2 diabetes and obesity. This provides further evidence for the appropriateness of the HIT control for clinical trials examining the effect of EFT on delay discounting in adults with self-reported type 2 diabetes.

**Data Availability Statement:** Deidentified raw data, analysis scripts, and survey files are available in a public repository: https://github.com/jeremiahmbrown/public-remedi-hit-pilot.

**Funding:** This work was funded by NIH grant: R01DK129567, awarded to JSS. National Institute for Diabetes and Digestive and Kidney Diseases. https://www.niddk.nih.gov/ The funders had no role in study design, data collection and analysis, decision to publish, or preparation of the manuscript.

**Competing interests:** Brown: None. Bickel: Although the following activities/relationships do not create a conflict of interest pertaining to this article, in the interest of full disclosure, Warren K. Bickel would like to report the following: Warren K. Bickel is a principal of HealthSim, LLC; BEAM Diagnostics, Inc.; and Red 5 Group, LLC. In addition, he serves on the scientific advisory board for Sober Grid, Inc.; and Ria Health; serves as a consultant for Boehringer Ingelheim International; and works on a project supported by Indivior, Inc. Epstein: None. Stein: None. This does not alter our adherence to PLOS ONE policies on sharing data and materials.

## Introduction

Episodic future thinking (EFT), the vivid imagining of personal future events, has been shown to reliably reduce delay discounting, the devaluation of future rewards [1, 2]. Higher delay discounting rates are associated with poorer glycemic control and diabetes-related self-care [3]. Importantly, a change in delay discounting over one year was predictive of increased HbA1c (glycated hemoglobin, a measure of retrospective blood sugar within the last 3 months) in adults with prediabetes, while controlling for demographic variables and body mass index (BMI; [4]). In recent years, researchers have examined the viability of EFT as an intervention to decrease delay discounting and increase engagement in health-promoting behaviors. In basic laboratory arrangements, EFT has been shown to reduce caloric consumption [5, 6], calories purchased in an online grocery shopping task [7], and behavioral economic demand for fast food [8]. EFT has been similarly efficacious in naturalistic settings, reducing caloric consumption [9] and calories mothers purchase while grocery shopping [10]. Finally, in clinical settings, EFT has been shown to reduce alcohol consumption [11] and increase medication adherence in adults with type 2 diabetes mellitus (T2DM; [12]); however, at least one other translational effort to incorporate EFT as a component of a clinical intervention has been less successful [13].

Regardless of the translation setting or phase, studies examining EFT's effects typically include a control condition for comparison. A common control condition in basic laboratory settings is episodic-recent thinking (ERT), the vivid remembering of recent personal events (i.e., within the past few days). While ERT is an appropriate control in the laboratory or naturalistic settings, repeated (i.e., daily or multiple times per day) engagement in ERT over weeks or months is impractical in clinical settings [14]. To keep cues related to recent events, ERT cues would need to be generated often; matching EFT cue generation to this schedule would be prohibitive. Additionally, control conditions in clinical experiments are more likely to promote adherence to the intervention and minimize demand characteristics if they equate expectation of improvement between groups. Thus, participants should view the control condition as clinically relevant or potentially helpful as the active participants view the potential helpfulness of the intervention. However, participants engaging in ERT do not report the belief that ERT will increase the likelihood of engagement in health-promoting behaviors or decrease delay discounting [14], which may limit adherence in clinical trials.

To address these limitations, Rung and Epstein [14] developed Health Information Thinking (HIT), a novel control condition well suited for long-term investigations on the effects of EFT on delay discounting and other health behaviors in clinical settings. In an online experiment, Rung and Epstein [14] demonstrated that participants (a general sample of 254 Amazon Mechanical Turk workers] engaging in EFT had significantly lower delay discounting than participants engaging in ERT or HIT. In the HIT control, participants read and responded to health-information vignettes and generated HIT cues by describing their reactions. HIT participants also rated the information on several dimensions, including how much they liked learning the information, the importance of understanding the information, how exciting it was to learn the information, and how useful it was to learn the information. This approach results in personalized summaries of and reactions to health information, which mirrors common survey-guided EFT cue generation methods [15]. Additionally, the health-information vignettes may be tailored for a given clinical condition, increasing the likelihood that individuals with said condition will view HIT cues as a relevant intervention component. Thus, the HIT condition is likely to control for participants' expectation of improvement and allows for matching cue generation and regeneration schedules, making HIT a potentially powerful control condition for long-term clinical studies examining the effect of EFT on delay discounting and health behaviors.

Although Rung and Epstein [14] showed that the HIT condition did not significantly impact delay discounting, the authors recruited participants from the general population and provided health information about a broad range of topics (e.g., effects of alcohol on sleep, recognizing symptoms of depression, understanding nutrition labels). In contrast, studies exploring the clinical effects of EFT have primarily focused on specific health behaviors in individual clinical populations (e.g., diet and physical activity in patients with prediabetes). It is unknown if a modified form of the HIT condition in which the health information is relevant to the prevention or treatment of a specific clinical target and population (e.g., weight loss and glycemic control in patients with T2DM) would yield different conclusions. Specifically, individuals with T2DM and obesity who generate HIT cues based on T2DM-related health-information vignettes (e.g., self-monitoring food, aerobic activity, nutrition labels) may react to these cues with increased motivation to engage in health-promoting behaviors, potentially spurring prospection and reducing delay discounting.

To further develop HIT as a clinical control for this population, the present study describes an online experiment in which participants with self-reported T2DM and obesity were recruited using Amazon Mechanical Turk and randomized to EFT, HIT, or a no-cue control (NCC) condition. EFT and HIT participants generated cues using a self-guided survey, while NCC participants did not generate cues. Afterward, participants completed a delay discounting task while instructed to vividly imagine or consider their EFT or HIT cues; NCC participants completed the delay discounting task without these instructions. We hypothesized that participants who were instructed to engage in EFT during the delay discounting task would have lower levels of delay discounting (i.e., more frequent selection of larger later rewards) compared to both HIT and NCC participants.

## Materials and methods

### Participants

We recruited 434 participants with self-reported obesity and T2DM using Amazon Mechanical Turk, from February 10th, 2022, to November 17th, 2022. The qualification criteria in Amazon Mechanical Turk specified that participants must be located in the US, have completed at least 100 previous human-intelligence tasks, and have an approval rating of at least 99%. To prevent individuals outside of the US from accessing the eligibility screening block or attempting to screen multiple times, we applied three sequential screening criteria to all responses. First, we used IP address screening to prevent participants using a virtual proxy network (VPN) from accessing the screening block [16]. Secondly, we prevented participants with IP addresses associated with countries outside of the US from accessing the screening block. Thirdly, we prevented participants with IP addresses that had already attempted or completed the survey from accessing the survey a second time. Participants who were not flagged for using a VPN, an IP address outside of the US, or a duplicate IP address were then shown the eligibility screening block. To be eligible, participants must have self-reported height and weight corresponding with a BMI $\geq$ 30 and self-reported a diagnosis of T2DM. To increase confidence in the truthfulness of self-reported T2DM, we asked participants to indicate any medical conditions (diagnosed by a medical professional) using two separate medical diagnoses questions. Each diagnosis question included the same 13 answer choices (i.e., various diseases and medical conditions; see survey file hosted on GitHub [https://github.com/jeremiahmbrown/public-remedi-hit-pilot] for complete list) presented in random order. After answering the first diagnosis question, participants answered three other unrelated questions (i.e., height, weight, and age) before answering the second diagnosis question on a separate page. To continue to the informed consent block, the selected answer options in both

diagnoses questions had to match, and participants could not have selected more than 4 total diagnoses on either diagnoses questions. Additionally, we prevented participants from continuing if they selected any of the following answer choices on either diagnosis question: Type 1 diabetes, chronic obstructive pulmonary disease (COPD), colorectal cancer, breast cancer, or none of the illnesses in this list. The eligibility criteria regarding BMI, T2DM, and other medical diagnoses were assessed at the same stage in the survey; violating one or more of the criteria rendered participants ineligible and ended the survey. Fig 1 depicts a flow diagram of participant enrollment, eligibility, dropout, and completion.

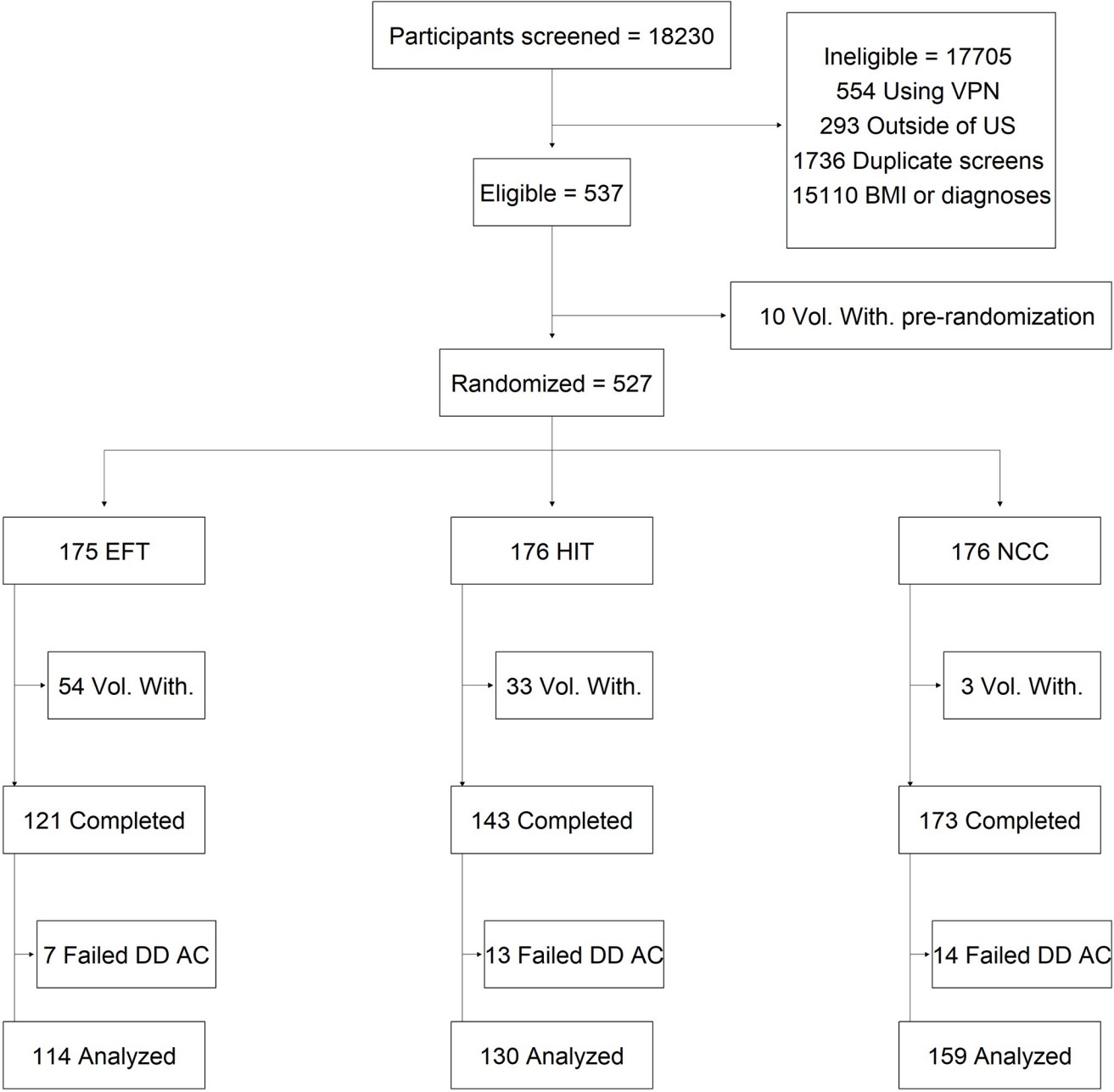

**Fig 1. Participant flow diagram.** Flow diagram depicting screening of MTurkers, assessment of eligibility, randomization, and study completion. Vol. With. = Voluntary withdrawal (i.e., did not continue responding to the survey).

We determined the target sample size using an a priori power analysis. 396 participants (132 per group) provide 95% power to detect a medium effect size difference (Cohen's $d$ = 0.5) in delay discounting between individual pairwise comparisons (e.g., EFT vs. HIT; EFT vs. NCC) following an omnibus ANOVA. We assumed a medium effect size based on values observed by Rung and Epstein [14]. Higher attrition rates were observed in the EFT group than in the HIT and NCC control groups; we discontinued recruitment after obtaining 434 completed responses, resulting in 172 completed responses in the NCC control group, 142 in the HIT group, and 120 in the EFT group.

## Procedure

This research was approved by the Virginia Tech IRB and performed in accordance with the 1964 Declaration of Helsinki. Participants completed the experiment online using Qualtrics survey software. After assessing eligibility as described in the Participants Section, participants provided informed consent by reading an IRB-approved consent information sheet and indicating their agreement; acquiring written informed consent was waived by the IRB. Participants then completed a demographic questionnaire before being randomized to EFT, HIT, or NCC conditions. EFT and HIT participants then generated seven cues according to their condition (participants assigned to the NCC condition did not generate cues) before completing an adjusting-amount delay discounting task [17]. After completing the delay discounting task, participants were thanked for their time and compensated $2.50 via Amazon Mechanical Turk. Additionally, participants could earn a $2.50 bonus payment if participants passed three of four attention checks embedded in the adjusting-amount delay discounting task and generated cues longer than 50 characters (if randomized to the EFT or HIT conditions).

**Demographics questionnaire.** Participants answered questions regarding their age, income, gender, and other sociodemographic characteristics. Additionally, participants reported their most recent HbA1c reading and indicated their current thoughts and actions regarding losing weight and better managing diabetes via a contemplation ladder [18]. The ladder ranged between zero and ten, with zero representing "I have no thoughts or plans to lose weight and better manage my blood sugar" and ten representing "I am taking actions to lose weight and better manage my blood sugar".

**EFT cue generation.** Participants randomized to the EFT condition generated EFT cues using a self-guided cue generation survey (see Brown & Stein [15], for details on this method). Participants first identified personally meaningful possible future events using single sentence descriptions (e.g., "In one month, I am celebrating my wife's birthday"), then rated those events on four dimensions: how much they enjoy the event, how important the event is, how exciting the event is, and how vividly they can imagine experiencing the event. Afterward, participants generated additional vivid, episodic details describing that event, creating a future thinking cue. For example, a completed EFT cue may read,

> In one month, I am celebrating my wife's birthday. We are having a nice dinner with friends and family at a local restaurant. I am asking her about her year and what she is looking forward to this upcoming year. I am happy to be surrounded by friends and family.

Participants generated seven EFT cues across seven future time frames: one month, three months, six months, one year, three years, five years, and ten years. These time frames matched the time frames used in the adjusting-amount delay discounting task.

**HIT cue generation.** Participants randomized to the HIT condition generated seven cues based on T2DM-related health-information vignettes; specifically: aerobic physical activity,

ultra-processed foods, self-monitoring, glycemic index, energy density, variety (of food), and strengthening exercises. Participants first read the vignette, then described (using a single sentence) one specific piece of information learned while reading. Afterward, participants rated their description on four dimensions: how much they liked learning the information, how important it was to learn the information, how exciting it was to learn the information, and how useful it was to learn the information. This process was repeated for the remaining vignettes. To complete the HIT cues, participants reread the vignettes and provided additional details to their original descriptions. Participants were prompted to include details regarding how the information fits into their existing knowledge, what the information made them think about, who or what the information might be useful for, and how the information made them feel. For example, a complete HIT cue generated after reading the vignette about strengthening exercises may read,

> I learned that strength training can help improve blood sugar level and decrease fat mass, along with several other benefits. I already knew many of the benefits of strength training, but this reinforced several concepts. The information made me think about my own strength training schedule. The information is useful for anyone who may not know how or why to start strength training. The information reinforced my love for strength training exercises

Examples of HIT cues generated from other health-information vignettes can be found in the supplementary materials.

**Delay discounting task.** Participants completed an adjusting-amount delay discounting task for a $1000 (USD) gain, including seven-time frames displayed in random order: one month, three months, six months, one year, three years, five years, and ten years [17]. Participants indicated their preferences between a smaller monetary reward available immediately or a larger one available after a delay. Depending on participants' choices, the amount of the smaller reward was titrated upwards or downwards after each trial; participants made a total of six choices within each block. Four attention checks were embedded into the task, asking participants to choose between receiving $500 now and $1000 now; these checks were the final choices during the one-month, six-months, three-years, and ten-years blocks. During the discounting task, EFT participants were instructed to engage in EFT, while HIT participants were instructed to engage in HIT; NCC participants completed the delay discounting task without additional instructions.

*EFT-cued delay discounting.* To elicit EFT during the discounting task, EFT participants were shown one EFT cue before beginning one block of choice trials in the discounting task. Additionally, the single-sentence descriptions for each cue were presented during each trial within a block. Participants were instructed to indicate their preferred option while imagining their single-sentence description. EFT cue time frames were matched with the delay discounting time frames (i.e., participants engaged with their one-month EFT cue while completing the one-month block in the discounting task).

*HIT-cued delay discounting.* To elicit HIT during the discounting task, HIT participants were shown one complete HIT cue before beginning one block of choice trials in the discounting task. Additionally, the single sentence description of the specific piece of information first identified in the HIT cue generation process (e.g., "I learned that muscle strengthening is recommended for those suffering from diabetes") was presented during each trial within the block. Participants were instructed to indicate their preferred option while imagining their single-sentence description. As HIT cues do not involve future time frames, cues were matched based on the order of appearance; the HIT cue generated first was paired with the first future

time frame in the discounting task (i.e., participants were exposed to aerobic physical activity cues while completing the one month block).

## Data analysis

Data were accessed continually for research purposes from February 2022 to November 2022 to monitor study progress and for interim analyses, and from November 2022 to April 2023 to conduct final analyses reported in this manuscript. All data analyses were completed using R (version 4.2.1; [19]) and RStudio [20]. The *tidyverse* package was used to wrangle, clean, and shape data; the *gtsummary* package was used to create Table 1; the *ggplot2* package was used to create all figures [21–23]. We compared EFT and HIT participant's average ratings of cues to test for group differences in liking/enjoyment, importance, and excitement using three t-tests (i.e., one for each cue characteristic; ratings of usefulness generated by HIT participants and ratings of vividness generated by EFT ratings could not be compared). We controlled for type I error rate using Bonferroni correction ($\alpha$ = .0167). To quantify delay discounting, ordinal area under the curve (AUC) values were calculated using the *discAUC* package [24, 25]. Higher values of AUC reflect lower levels of delay discounting. To assess if delay discounting varied between EFT, HIT, and NCC groups, we performed a one-way ANOVA with Tukey HSD post hoc comparisons. As a sensitivity analysis, we performed these tests twice: once including only ordinal AUC values from participants who correctly answered three out of four attention checks during the delay discounting task, and again including all ordinal AUC values (see supporting information). To test for differences in delay discounting attention check passing rates

**Table 1. Demographics of participants who completed all study procedures, overall and by group assignment.**

| Variable | Overall | Group | | | p-value[2] |
| --- | --- | --- | --- | --- | --- |
| | | HIT | EFT | NCC | |
| | N = 434[1] | N = 142[1] | N = 120[1] | N = 172[1] | |
| **Age (years)** | 44.00 (35.00, 53.75) | 43.00 (34.00, 51.00) | 43.50 (35.00, 54.00) | 45.00 (36.00, 55.00) | 0.4 |
| **BMI** | 37.12 (33.40, 43.31) | 36.75 (32.46, 43.67) | 36.86 (33.45, 42.00) | 37.90 (34.12, 43.90) | 0.5 |
| **Income (USD)** | 34,999.50 (4,999.50, 54,999.50) | 34,999.50 (4,999.50, 54,999.50) | 34,999.50 (4,999.50, 54,999.50) | 34,999.50 (4,999.50, 54,999.50) | 0.6 |
| Unknown | 11 | 6 | 2 | 3 | |
| **Contemplation Ladder** | 9.00 (8.00, 10.00) | 9.00 (8.00, 10.00) | 9.00 (8.00, 10.00) | 9.00 (8.00, 10.00) | 0.7 |
| Unknown | 4 | 1 | 1 | 2 | |
| **HbA1c** | | | | | 0.8 |
| 6.9% or lower | 168 / 434 (39%) | 61 / 142 (43%) | 41 / 120 (34%) | 66 / 172 (38%) | |
| 7.0%–8.0% | 125 / 434 (29%) | 35 / 142 (25%) | 41 / 120 (34%) | 49 / 172 (28%) | |
| 8.0%–8.9% | 46 / 434 (11%) | 16 / 142 (11%) | 11 / 120 (9.2%) | 19 / 172 (11%) | |
| 9.0% or greater | 38 / 434 (8.8%) | 11 / 142 (7.7%) | 10 / 120 (8.3%) | 17 / 172 (9.9%) | |
| Unknown by participant | 57 / 434 (13%) | 19 / 142 (13%) | 17 / 120 (14%) | 21 / 172 (12%) | |
| **Gender** | | | | | 0.6 |
| Female | 269 / 434 (62%) | 91 / 142 (64%) | 69 / 120 (57%) | 109 / 172 (63%) | |
| Male | 162 / 434 (37%) | 51 / 142 (36%) | 50 / 120 (42%) | 61 / 172 (35%) | |
| Other | 3 / 434 (0.7%) | 0 / 142 (0%) | 1 / 120 (0.8%) | 2 / 172 (1.2%) | |

**The income** variable has been recoded as a continuous variable; see note in data analysis section.

[1]Median (IQR); n / N (%)

[2]Kruskal-Wallis rank sum test; Pearson's Chi-squared test; Fisher's exact test

between groups, we used a logistic regression to predict attention check passing status as a function of group. To test for differences in study attrition between groups, we used Fisher's exact test. We used a significance level of $\alpha$ = 0.05 to determine statistical significance for all tests, unless otherwise stated. Income was self-reported using an ordinal scale (e.g., $10,000–19,999); in order to display income data in Table 1, income was recoded as a continuous variable by assigning participants a value in the center of their selected category (e.g., participants who reported $10,000 - $19,999 were recoded as $14,999.50). Deidentified raw data and R code used to generate results are available on the authors' Github page (https://github.com/jeremiahmbrown/public-remedi-hit-pilot).

## Results

### Demographics

Table 1 depicts demographic information of participants who completed the experiment; groups were not significantly different in regards to age, BMI, income, contemplation ladder score, self-reported HbA1c, and gender.

### Cue ratings

Cue ratings for the liking/enjoyment characteristic were significantly higher for EFT participants (M = 4.59; SE = .025) than HIT participants (M = 3.89, SE = .036), $t(219.56)$ = 8.73, $p$ = < .001. Cue ratings for the importance characteristic were not significantly different between EFT participants (M = 4.29, SE = .036) and HIT participants (M = 4.25, SE = .031), $t(259.84)$ = 0.18, $p$ = .85. Cue ratings for the excitement characteristic were significantly higher for EFT participants (M = 4.33, SE = .032) than HIT participants (M = 3.28, SE = .04), $t(226.69)$ = 10.87, $p$ = < .001. A bar chart depicting mean and SE of average participant ratings by group across characteristics is available in the supplementary materials.

### Delay discounting

Fig 2 depicts the decline in the subjective value of the larger later reward relative to the smaller sooner reward (i.e., indifference points) as a function of delay across EFT, HIT, and NCC groups. Fig 3 depicts a notched boxplot of corresponding ordinal AUC values in these groups. Data in Figs 2 and 3 includes participants who passed three or more discounting attention checks (see supporting information for recreations of Figs 2 and 3, including all participants who completed the experiment). Results of a logistic regression predicting delay discounting attention check passing rates between groups revealed no significant effect of group; a model including group as a predictor of passing rates was not significantly different than the null model, $X^2$ = 1.04, df = 2, $p$ = .60. Results of the one-way ANOVA examining differences in ordinal AUC between groups (including participants who passed three or more discounting attention checks; $n$ = 113, $n$ = 129, and $n$ = 159 for EFT, HIT, and NCC groups, respectively) indicate that at least one group generated significantly different AUC values ($F(2, 398)$ = 8.54, $p < .001$). Post hoc comparisons using Tukey's HSD indicate that the EFT group demonstrated significantly higher ordinal AUC values than the HIT group (Mean difference = 0.103, 95% CI: 0.04–0.167, $p < .001$, Cohen's $d$ = .49, 95% CI: 0.23–0.75) and the NCC group (mean difference = 0.089, 95% CI: 0.0279–0.149, $p$ = .002, Cohen's $d$ = .43, 95% CI: 0.18–0.67). Additionally, no difference between the NCC and the HIT group was observed (mean difference = 0.014, 95% CI: -0.042–0.072, $p$ = .81, Cohen's $d$ = .07, 95% CI: -0.16–0.30). All conclusions remained consistent when including all ordinal AUC values, regardless of

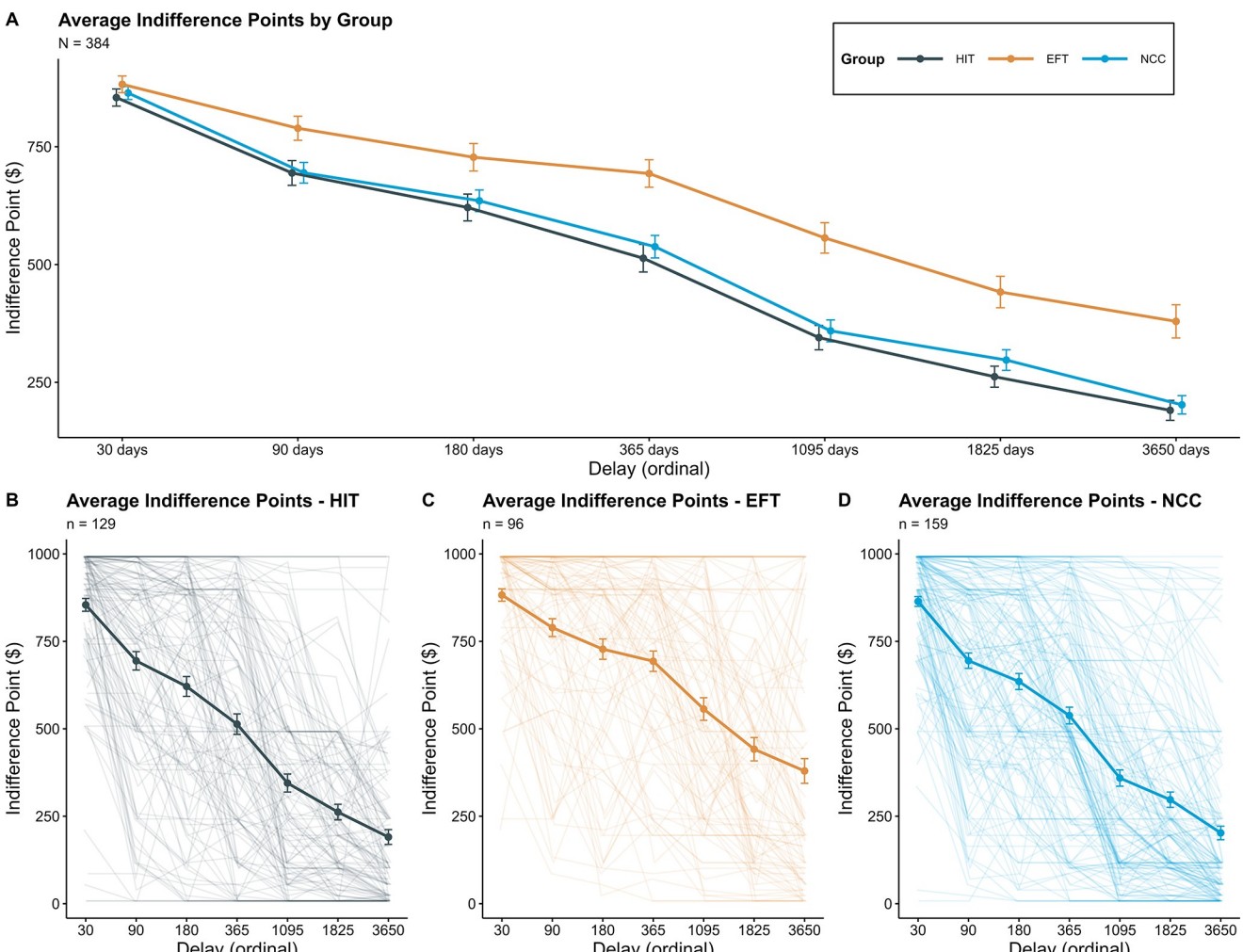

**Fig 2. Group-level mean and individual indifference points.** Mean and individual indifference points of participants who passed at least three of four attention checks during the discounting task. Panel A depicts mean indifference points as a function of delay by group assignment, plotted on an ordinal scale. Panels B, C, and D depict mean indifference points within each group (bold lines); transparent lines depict individual subject discounting curves. Error bars represent standard error.

successfully passing three out of four attention checks during the delay discounting task (see supporting information for model results).

## Attrition

As depicted in the flow diagram (Fig 1), many eligible participants discontinued after randomization (i.e., during cue generation or the discounting task that followed) and could, therefore, not be included in the analysis of delay discounting. Of the 174 randomized EFT participants, 120 (69%) completed the study (54 voluntarily withdrew after randomization). Of the 175 randomized HIT participants, 142 (81%) completed the study (33 voluntarily withdrew after randomization). Of the 175 randomized NCC participants, 172 (98%) completed the study (3 voluntarily withdrew after randomization). Fisher's exact test was used to determine if the proportion of participants randomized to each group compared to participants who completed

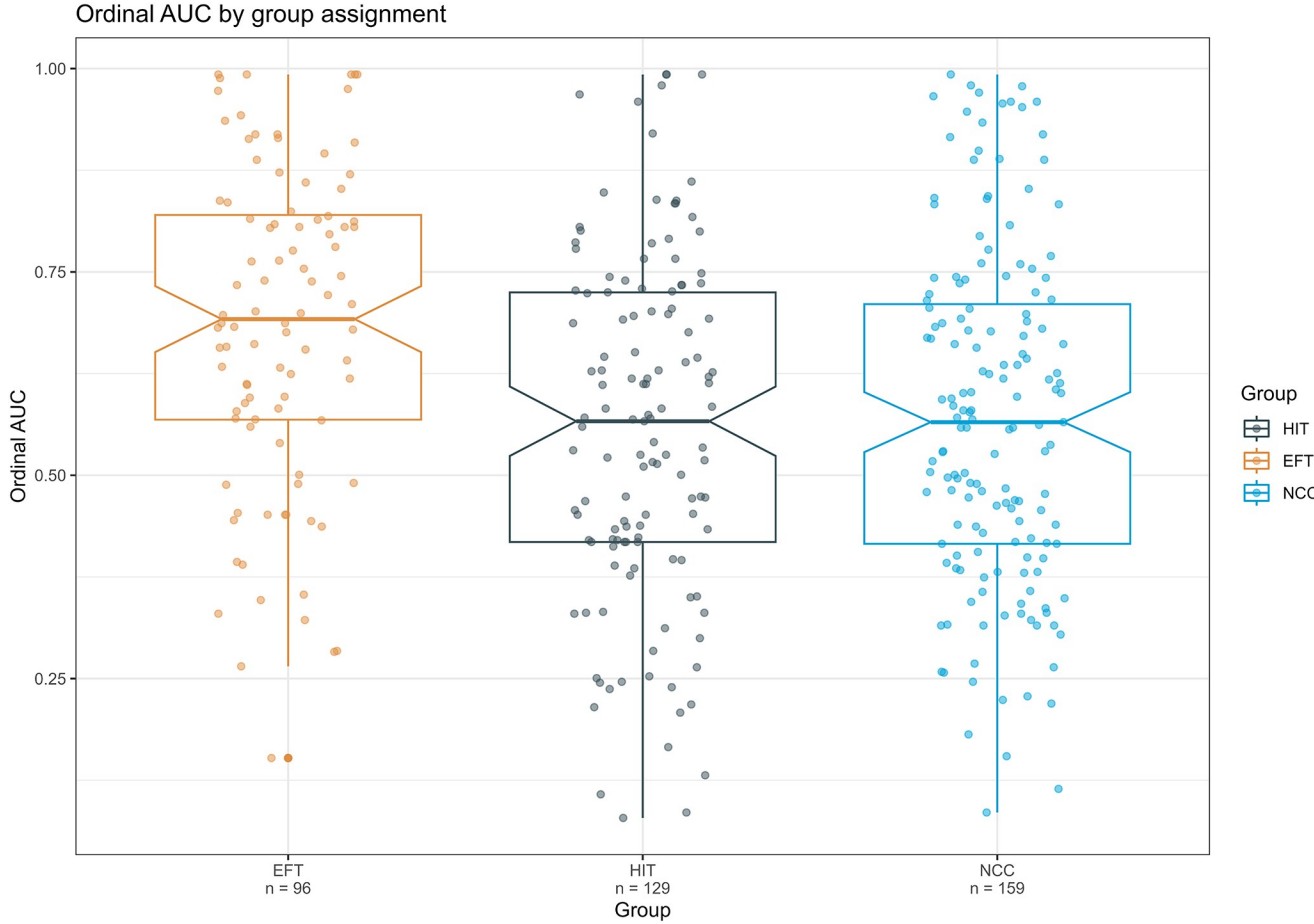

**Fig 3. Ordinal AUC by group.** Notched boxplot of grouped ordinal AUC values of participants who passed at least three of four attention checks during the discounting task. Notches represent the median $\pm$ 1.58 * IQR/$\sqrt{n}$.

the study in each group were different. Based on the $p$-value obtained from the test ($p < .001$), we reject the null hypothesis (i.e., that the number of participants who voluntarily withdrew vs. completed the study after randomization were the same between groups). Pairwise comparisons using Fisher's exact test revealed significant differences in attrition rates between all groups (EFT vs. HIT: $p = .002$; EFT vs. NCC: $p < .001$; HIT vs. NCC: $p = < .001$).

## Discussion

In this online experiment, we made two contributions to the literature: 1) the HIT control (using cues generated in response to T2DM related health-information) does not result in lower delay discounting compared to a no-cue control condition in adults with self-reported obesity and T2DM, and 2) brief engagement in EFT does result in lower delay discounting compared to the HIT control and a no-cue control in adults with self-reported obesity and T2DM. We will discuss both contributions, study limitations, and future directions.

Health Information Thinking appears to be a suitable control for clinical trials involving repeated engagement in EFT as an intervention to reduce delay discounting. In this clinical sample (adults with self-reported obesity and T2DM), engaging in HIT using cues generated

from T2DM-related health-information vignettes did not result in lower delay discounting relative to participants engaging in EFT. Participants engaging in EFT exhibited significantly greater ordinal AUC (i.e., lower delay discounting) than HIT and NCC conditions. No differences in ordinal AUC were observed between HIT and NCC participants, suggesting that HIT is an inert intervention concerning delay discounting. These results are consistent with Rung and Epstein [14], who showed that the HIT and ERT control conditions produced comparable levels of delay discounting. These findings have important implications; namely, that HIT does not appear to influence delay discounting, thus making it appropriate as a control condition in clinical studies designed to reduce delay discounting and influence health behavior. Likewise, the HIT condition is structurally similar to the EFT intervention, as the number of vignettes and cues can be matched to the future events used in EFT. This provides a strong control for effort and time in clinical studies, as participants in both the active intervention and control groups would be scheduled to complete cue generation at approximately the same frequency.

These results suggest that EFT may be an efficacious intervention to reduce delay discounting in adults with obesity and T2DM. To date, only one multiple-baseline small-*n* study has examined the effects of EFT on medication adherence and delay discounting in adults with T2DM [12]. This prior study reported a large effect size difference in discounting between baseline and EFT conditions (ES = 1.2), although the small sample size (n = 4) prevented inferential statistics. Thus, the present study's findings are important to the literature examining EFT as a clinical intervention for individuals with T2DM. However, as these findings are limited to acute (i.e, single exposure) EFT and delay discounting, future research should examine the effects of repeated (i.e., multiple exposures) EFT on delay discounting, health behaviors, and health outcomes.

Although the present study is the first large-scale examination of EFT in adults with self-reported T2DM, other studies have shown that acute and repeated EFT reliably reduces or results in lower delay discounting compared to control conditions in adults with prediabetes [13, 26, 27], a condition of elevated blood glucose that increases the risk of developing T2DM [28]. While EFT did not impact other health behavior outcomes in the aforementioned studies of adults with prediabetes, other work in adults with overweight or obesity has been promising [5, 8–10, 29]. Thus, more work is required to explore potential moderators of the efficacy of EFT (e.g., HbA1c, education, BMI) in regard to changing health behaviors.

This study is not without limitations that should be addressed in future research. First, while we are reasonably confident that our screening strategies limited the number of participants who completed the study but do not have a diagnosis of T2DM or obesity, it is likely some unknown number who did not meet our qualifications were able to meet the screening criteria by chance. However, this number is likely to be low, as we implemented emerging best practices for online research utilizing Amazon Mechanical Turk [16], and further developed methods to limit the ability to misrepresent a T2DM diagnosis in screening. As a result, less than 3% of participants could continue past the screening questionnaire. Additionally, previous research supports the validity of self-reported T2DM via interview methods, although it is unknown if these results generalize to self-reported T2DM via an online survey [30–32]. Furthermore, the prevalence of self-reported T2DM in MTurkers aged 40–49 (measured in 2015) was not different than the prevalence of T2DM in a representative sample of Americans (i.e., the 2013 Behavioral Risk Factor Surveillance System; [33]).

Second, we observed significantly higher attrition rates in participants randomized to the EFT group compared to participants randomized to HIT or NCC groups; this difference may be due to the increased effort required to generate EFT cues and engage in EFT during the delay discounting task. Importantly, no significant group differences in demographic variables related to delay discounting were observed (i.e., age, BMI, income, education, HbA1c), which

suggests that the observed differences in attrition did not introduce bias related to any measured participant characteristic. We note, however, this differential attrition may be less likely to occur in a sample of motivated participants enrolled in a clinical trial using behavioral interventions and EFT to manage T2DM better.

Thirdly, one other potential benefit of HIT was elucidated by Rung and Epstein [14] but was not explored in the present study; namely, the ability to control for participants' expectation of improvement. Future research should measure participants' expectations of improvement associated with EFT and HIT conditions. Fourthly, Rung and Epstein [14] and the present study were unable to examine the effects of HIT (acute or repeated) on other health behaviors (i.e., food consumption, physical activity engagement); future research is required to examine these outcomes in clinical samples and to compare expectations of improvement in the context of a clinical trial. Indeed, our group is currently using the HIT control condition in a clinical trial examining the effects of repeated engagement in EFT (embedded within a behavioral intervention) on change in delay discounting, body weight, and HbA1c in adults with type 2 diabetes and obesity (see https://clinicaltrials.gov/ct2/show/NCT05280925). While data collection is ongoing, observing significant group differences in behavioral outcomes would suggest that the HIT control is inert with regard to delay discounting, health behaviors, and health outcomes. Similarly, observing no differences in participants' treatment acceptability ratings and perceived helpfulness would suggest that expectations of improvement are not different between EFT and HIT.

Fifth, we observed significantly different average participant ratings of EFT and HIT cues on excitement and liking/enjoyment characteristics, although no differences in ratings of importance were observed. While differences in cue ratings between active and control groups are not ideal, EFT vs. HIT cue generation are, by design, different exercises resulting in qualitatively different written descriptions (i.e., a narrative about personal engagement in possible future events vs. a personal reaction to health information). Therefore, differences in EFT and HIT cues on ratings of excitement and liking/enjoyment may be related to the expected differences in content between EFT vs. HIT. Nonetheless, when applied in a long-term RCT involving repeated, daily engagement in EFT or HIT, these content differences may reduce adherence to HIT and, therefore, its suitability as a control condition. Future research should examine possible differences in intervention adherence between EFT and HIT in long-term RCTs.

In conclusion, the present experiment increases our confidence in using the HIT control for clinical examinations of the effect of EFT on delay discounting. The HIT control cues can be generated using health-information vignettes directly relevant to the clinical sample; engaging in HIT using these cues does not result in lower delay discounting relative to a standard delay discounting task. HIT control is a promising candidate for long-term clinical trials involving EFT as an intervention to reduce delay discounting and change health behaviors.

## Supporting information

**S1 Fig. Group-level mean and individual indifference points, all participants.** Mean and individual indifference points of participants, including all participants. Panel A depicts mean indifference points as a function of delay by group assignment, plotted on an ordinal scale. Panels B, C, and D depict mean indifference points within each group (bold lines); transparent lines depict individual subject discounting curves. Error bars represent standard error. (TIFF)

**S2 Fig. Ordinal AUC by group, all participants.** Notched boxplot of grouped ordinal AUC values, including all participants. Notches represent the median $\pm 1.58 * IQR/\sqrt{n}$. (TIFF)

**S3 Fig. Mean and SE of characteristic ratings for EFT and HIT cues.**
(PNG)

**S1 Table. Demographic table of study completers vs. non-completers.**
(PDF)

**S1 Appendix. Sensitivity analysis of AUC comparisons.**
(PDF)

## Author Contributions

**Conceptualization:** Jeremiah M. Brown, Warren K. Bickel, Leonard H. Epstein, Jeffrey S. Stein.

**Data curation:** Jeremiah M. Brown.

**Formal analysis:** Jeremiah M. Brown.

**Funding acquisition:** Warren K. Bickel, Leonard H. Epstein, Jeffrey S. Stein.

**Investigation:** Jeremiah M. Brown, Jeffrey S. Stein.

**Methodology:** Jeremiah M. Brown, Jeffrey S. Stein.

**Project administration:** Jeremiah M. Brown, Jeffrey S. Stein.

**Resources:** Jeffrey S. Stein.

**Supervision:** Jeffrey S. Stein.

**Visualization:** Jeremiah M. Brown.

**Writing – original draft:** Jeremiah M. Brown.

**Writing – review & editing:** Jeremiah M. Brown, Warren K. Bickel, Leonard H. Epstein, Jeffrey S. Stein.

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
