## [Decision Letter · Decision Letter 0]

6 Jun 2023

PONE-D-23-10564Episodic future thinking in type 2 diabetes: Further development and validation of the health information thinking control for clinical trialsPLOS ONE

Dear Dr. Stein,

Thank you for submitting your manuscript to PLOS ONE. After careful consideration, we feel that it has merit but does not fully meet PLOS ONE’s publication criteria as it currently stands. Therefore, we invite you to submit a revised version of the manuscript that addresses the points raised during the review process.

ACADEMIC EDITOR: I have reviewed your manuscript and the feedback from the expert reviewer. I agree with the reviewers' comments as well as their overall assessment of the manuscript's strengths and likely worthiness for publication. Therefore, I invite you to attend the reviewer's minor suggested revisions and resubmit the manuscript. Thank you for submitting to PLOS ONE, and I look forward to seeing your revision soon.

We look forward to receiving your revised manuscript.

Kind regards,

Emily Lund

Academic Editor

PLOS ONE

“I have read the journal's policy and the authors of this manuscript have the following competing interests:

Brown: None.

Bickel: Although the following activities/relationships do not create a conflict of interest pertaining to this article, in the interest of full disclosure, Warren K. Bickel would like to report the following: Warren K. Bickel is a principal of HealthSim, LLC; BEAM Diagnostics, Inc.; and Red 5 Group, LLC. In addition, he serves on the scientific advisory board for Sober Grid, Inc.; and Ria Health; serves as a consultant for Boehringer Ingelheim International; and works on a project supported by Indivior, Inc.

Epstein: None.

Stein: None.”

5. Please include your tables as part of your main manuscript and remove the individual files. Please note that supplementary tables (should remain/ be uploaded) as separate "supporting information" files.

Reviewers' comments:

Reviewer's Responses to Questions

**Comments to the Author**

1. Is the manuscript technically sound, and do the data support the conclusions?

Reviewer #1: Yes

2. Has the statistical analysis been performed appropriately and rigorously? 

Reviewer #1: Yes

3. Have the authors made all data underlying the findings in their manuscript fully available?

Reviewer #1: Yes

4. Is the manuscript presented in an intelligible fashion and written in standard English?

Reviewer #1: Yes

5. Review Comments to the Author

Reviewer #1: I have reviewed the manuscript “Episodic future thinking in type 2 diabetes: Further development and validation of the health information thinking control for clinical trials” under consideration for publication in PLOS ONE. This manuscript explores Health Information Thinking (HIT) as a control condition for clinical trials examining the effect of Episodic Future Thinking (EFT) on delay discounting (DD) among adults with self-reported type 2 diabetes, which is likely to be of interest to a broader audience. Findings suggest that HIT could be a potential control condition for research examining the effects of EFT on DD, as individuals in the HIT condition had higher DD than individuals in the EFT condition. This manuscript addressed the limitation of previous studies on this topic that use general health information as HIT. I appreciate the fact that the authors utilized diabetes-specific HIT rather than general health information.

Below are some issues that I identified that could potentially improve a future manuscript (organized roughly by the appropriate section):

INTRODUCTION

1. Please provide brief explanations/definitions for technical terms (e.g., HbA1c, BMI) for readers who might not be familiar with these terms.

METHOD

2. I appreciate the authors providing a specific example of EFT Cue Generation, “In one month, I am celebrating my wife’s birthday. We are having a nice dinner with friends and family at a local restaurant. I am asking her about her year and what she is looking forward to this upcoming year. I am happy to be surrounded by friends and family,” on Page 9. I suggest the authors also provide a specific example of HIT Cue Generation to inform how the HIT cue generation might look like. Relatedly, readers might be interested in more details regarding each type of health information as HIT cues: aerobic physical activity, ultra-processed foods, self-monitoring, glycemic index, energy density, variety (of food), and strengthening exercises. I recognize additional information is provided in the supplemental materials, but if there is space, authors might provide an example generated cue for each HIT cue in the manuscript itself.

3. This may be better suited for the results section, but clearer presentation of the proportion of individuals who were considered valide responders (i.e., passed 3 of 4 attention checks) is needed. I recommend updating the flow chart (Figure 1) to reflect number of individuals who failed attention checks, as well as the final analyzed sample sizes as results presented are for those who both completed and passed attention checks (although the authors do mention similar results were observed for the full completer sample and provide those results in supplmental material).

4. I appreciate the authors conducting attrition analyses. I realize this may not be possible, but given the demographic items were administered at the start of the survey, are there data for enough individuals to allow for comparison of completers vs. non-completers to see if any baseline factors were associated with completion status?

5. Related to point 4, please indicate whether rates of valid responding (i.e., 3+ of 4 attention checks) differed by condition. Considering completers only, it appears to be 129/142 (90.8%) for HIT, 96/120 (80%) for EFT, 159/172 (92.4%) for NCC using info from the Delay Discounting Results section, but it is unclear whether these are meaningfully different rates. Please report this information and discuss as relevant. It could also be helpful to examine whether baseline factors were associated with responding status. Results could be helpful both in terms of: 1) establishing internal validity of the study and 2) worthy of discussion in terms of implications for intervention contexts (i.e., intervention adherence)

RESULTS/TABLES

6. The study asked participants to rate the EFT and HIT cues on four dimensions (i.e., enjoyment/likeliness, importance, excitement, vividness/usefulness). However, no data analyses were performed for these measurements. I recognize this is beyond the scope of the manuscript as currently written, but I am curious to learn if any dimension ratings are associated with DD rates, and imagine other readers might be interested as well. Or if there is any difference in enjoyment/likeliness, importance, and excitement observed across EFT and HIT conditions. If there are any differences (or similarly, limited differences) in these dimensions across conditions, this could provide further discussion of the suitability of HIT as a control condition for EFT studies.

7. This is a minor issue, but it could be helpful to report findings for the experimental conditions in the same order throughout the manuscript. For example, in the text, it is reported as EFT, HIT, NCC. Same for Figure 3. However, Figures 1 and 2 order as HIT, EFT, NCC.

DISCUSSION

8. This study measured DD once and then compared the group differences in DD, so no change in DD was measured. Thus, authors should be careful when using terms inferring change throughout the discussion (and also the abstract) to prevent inadvertently misleading readers. For example, in the abstract the authors indicate “that EFT, but not diabetes-specific HIT, reduces delay discounting in adults with type 2 diabetes and obesity”, but the results only indicated DD was lower in the EFT condition than in the HIT condition. Please revise as necessary accordingly

9. The first sentence of the discussion (i.e. that HIT does not result in lower DD relative to EFT) seems at odds with findings. Should this read “…does not significantly reduce delay discounting relative to either EFT or a NCC…”? Please clarify/correct as appropriate.

10. In the Limitations paragraph on page 17, the authors state there was greater attrition in the EFT group, but it is not clear from analyses reported whether this is accurate, only that there were differences among the 3 groups. Was attrition for EFT in fact different from HIT as a pairwise comparison?

11. In paragraph 4 of the Discussion session (page 16), the sentence “Thus, more work is required to explore demographic variables that may moderate the efficacy of EFT in regard to changing health behaviors” seems out of place in this paragraph. It seems more information is needed; for example are the authors referring to diabetes-related health status specifically? I suggest the authors elaborate more and provide examples of demographic variables they think might be potential moderators; otherwise, they could revise this paragraph and remove this sentence, as they did not discuss any potential moderators in the study analyses.

6. PLOS authors have the option to publish the peer review history of their article (what does this mean?). If published, this will include your full peer review and any attached files.

Reviewer #1: No

---

## [Author Response · Author response to Decision Letter 0]

14 Jul 2023

We have added additional details to the Procedure subsection of the Methods section regarding informed consent. Additionally, we have removed the Ethics Approval and Consent to Participate sections from the front matter of the manuscript. 

“I have read the journal's policy and the authors of this manuscript have the following competing interests:

Brown: None.

Bickel: Although the following activities/relationships do not create a conflict of interest pertaining to this article, in the interest of full disclosure, Warren K. Bickel would like to report the following: Warren K. Bickel is a principal of HealthSim, LLC; BEAM Diagnostics, Inc.; and Red 5 Group, LLC. In addition, he serves on the scientific advisory board for Sober Grid, Inc.; and Ria Health; serves as a consultant for Boehringer Ingelheim International; and works on a project supported by Indivior, Inc.

Epstein: None.

Stein: None.”

This information has now been included in the Competing Interest section of the front matter. 

We have made the above-described changes to the methods section.

5. Please include your tables as part of your main manuscript and remove the individual files. Please note that supplementary tables (should remain/ be uploaded) as separate "supporting information" files.

We have reviewed the reference list to ensure that it is complete and correct; no changes have been made, beyond additional citations included in revisions.

Review Comments to the Author

Reviewer #1: I have reviewed the manuscript “Episodic future thinking in type 2 diabetes: Further development and validation of the health information thinking control for clinical trials” under consideration for publication in PLOS ONE. This manuscript explores Health Information Thinking (HIT) as a control condition for clinical trials examining the effect of Episodic Future Thinking (EFT) on delay discounting (DD) among adults with self-reported type 2 diabetes, which is likely to be of interest to a broader audience. Findings suggest that HIT could be a potential control condition for research examining the effects of EFT on DD, as individuals in the HIT condition had higher DD than individuals in the EFT condition. This manuscript addressed the limitation of previous studies on this topic that use general health information as HIT. I appreciate the fact that the authors utilized diabetes-specific HIT rather than general health information.

Below are some issues that I identified that could potentially improve a future manuscript (organized roughly by the appropriate section):

INTRODUCTION

1. Please provide brief explanations/definitions for technical terms (e.g., HbA1c, BMI) for readers who might not be familiar with these terms.

These terms have now been defined in the introduction. 

METHOD

2. I appreciate the authors providing a specific example of EFT Cue Generation, “In one month, I am celebrating my wife’s birthday. We are having a nice dinner with friends and family at a local restaurant. I am asking her about her year and what she is looking forward to this upcoming year. I am happy to be surrounded by friends and family,” on Page 9. I suggest the authors also provide a specific example of HIT Cue Generation to inform how the HIT cue generation might look like. Relatedly, readers might be interested in more details regarding each type of health information as HIT cues: aerobic physical activity, ultra-processed foods, self-monitoring, glycemic index, energy density, variety (of food), and strengthening exercises. I recognize additional information is provided in the supplemental materials, but if there is space, authors might provide an example generated cue for each HIT cue in the manuscript itself.

We agree that an example of an HIT cue makes a good addition to the manuscript. As you observed, there are concerns regarding space if we were to include 6 more HIT cues (one for each health-information vignette). We added a sentence in this section reminding readers that examples of HIT cues related to each vignette are available in the supplemental materials.

3. This may be better suited for the results section, but clearer presentation of the proportion of individuals who were considered valid responders (i.e., passed 3 of 4 attention checks) is needed. I recommend updating the flow chart (Figure 1) to reflect number of individuals who failed attention checks, as well as the final analyzed sample sizes as results presented are for those who both completed and passed attention checks (although the authors do mention similar results were observed for the full completer sample and provide those results in supplemental material).

This is valuable feedback which has noticeably improved figure 1. We updated the flow chart (figure 1) to reflect the number of individuals who failed attention checks, as well as the final analyzed sample. We also rearranged the order of the groups, consistent with the feedback in comment 7.

4. I appreciate the authors conducting attrition analyses. I realize this may not be possible, but given the demographic items were administered at the start of the survey, are there data for enough individuals to allow for comparison of completers vs. non-completers to see if any baseline factors were associated with completion status?

Thanks for this idea. We agree that this is worth investigating. We have added an additional demographic table to the supplemental materials, depicting differences in demographic variables between study completers and noncompleters, all of whom were randomized to one of the three groups. The only difference between completers and noncompleters were found in regards to group assignment (as indicated in the attrition subsection of the results section).

5. Related to point 4, please indicate whether rates of valid responding (i.e., 3+ of 4 attention checks) differed by condition. Considering completers only, it appears to be 129/142 (90.8%) for HIT, 96/120 (80%) for EFT, 159/172 (92.4%) for NCC using info from the Delay Discounting Results section, but it is unclear whether these are meaningfully different rates. Please report this information and discuss as relevant. It could also be helpful to examine whether baseline factors were associated with responding status. Results could be helpful both in terms of: 1) establishing internal validity of the study and 2) worthy of discussion in terms of implications for intervention contexts (i.e., intervention adherence)

We thank the reviewer for this observation. After more thoroughly investigating the number of correctly answered DD attention checks between groups, we realized that our code was not correctly counting the attention check scores in the EFT group. We corrected this mistake; instead of 21 EFT participants failing the attention check criteria, only 7 did. The mistake was caused by an incorrect embedded data variable calculation in Qualtrics; to correct for this, we have now recalculated the attention check values for the EFT group using the raw question data in the R script. We then compare delay discounting attention check passing rates between groups using a Logistic regression; a model including group as a predictor of passing rates was not significantly different than a null model. Additionally, we have updated all analyses and figures in the manuscript to reflect the changes to the EFT group; the results of all the hypotheses testing were not changed, although there are small changes in relevant test statistics, p-values, and effect sizes. Note that these changes decreased the effect size when comparing the differences in the AUC observed in the EFT group to the HIT and NCC groups. 

RESULTS/TABLES

6. The study asked participants to rate the EFT and HIT cues on four dimensions (i.e., enjoyment/likeliness, importance, excitement, vividness/usefulness). However, no data analyses were performed for these measurements. I recognize this is beyond the scope of the manuscript as currently written, but I am curious to learn if any dimension ratings are associated with DD rates, and imagine other readers might be interested as well. Or if there is any difference in enjoyment/likeliness, importance, and excitement observed across EFT and HIT conditions. If there are any differences (or similarly, limited differences) in these dimensions across conditions, this could provide further discussion of the suitability of HIT as a control condition for EFT studies.

We agree that comparisons of cue ratings between EFT and HIT groups make for a valuable addition to the manuscript. We have edited the manuscript to include comparisons of cue ratings for the excitement, importance, and liking categories (vividness/usefulness can’t be directly compared across EFT and HIT tasks); these changes are reflected in the data analysis, results, and discussion. Additionally, we have included in the supplementary materials a figure depicting mean and SEs for characteristic ratings for EFT and HIT cues. We did not examine relationships between DD and cue ratings, as any analyses would exclude one third of our participants who did not generate EFT or HIT cues (i.e., the NCC participants).

7. This is a minor issue, but it could be helpful to report findings for the experimental conditions in the same order throughout the manuscript. For example, in the text, it is reported as EFT, HIT, NCC. Same for Figure 3. However, Figures 1 and 2 order as HIT, EFT, NCC.

Thanks for pointing this out. While it’s a minor issue, it’s simple to correct and makes the manuscript cleaner. We’ve corrected the figures to ensure that they all display group information in the same order. 

DISCUSSION

8. This study measured DD once and then compared the group differences in DD, so no change in DD was measured. Thus, authors should be careful when using terms inferring change throughout the discussion (and also the abstract) to prevent inadvertently misleading readers. For example, in the abstract the authors indicate “that EFT, but not diabetes-specific HIT, reduces delay discounting in adults with type 2 diabetes and obesity”, but the results only indicated DD was lower in the EFT condition than in the HIT condition. Please revise as necessary accordingly

Thanks for catching this. We have revised our language to avoid misleading readers.

9. The first sentence of the discussion (i.e. that HIT does not result in lower DD relative to EFT) seems at odds with findings. Should this read “…does not significantly reduce delay discounting relative to either EFT or a NCC…”? Please clarify/correct as appropriate.

Thanks for this suggestion. We rewrote this sentence in an attempt to increase clarity for readers.

10. In the Limitations paragraph on page 17, the authors state there was greater attrition in the EFT group, but it is not clear from analyses reported whether this is accurate, only that there were differences among the 3 groups. Was attrition for EFT in fact different from HIT as a pairwise comparison?

Thanks for this feedback. We now report the results of pairwise Fisher’s Exact Tests to examine pairwise differences between all three conditions; indeed, the rates of attrition were significantly different between all three groups. 

11. In paragraph 4 of the Discussion session (page 16), the sentence “Thus, more work is required to explore demographic variables that may moderate the efficacy of EFT in regard to changing health behaviors” seems out of place in this paragraph. It seems more information is needed; for example are the authors referring to diabetes-related health status specifically? I suggest the authors elaborate more and provide examples of demographic variables they think might be potential moderators; otherwise, they could revise this paragraph and remove this sentence, as they did not discuss any potential moderators in the study analyses.

We have rewritten this sentence to include examples of potentially moderating demographic characteristics.

---

## [Editor Report · Decision Letter 1]

20 Jul 2023

Episodic future thinking in type 2 diabetes: Further development and validation of the health information thinking control for clinical trials

PONE-D-23-10564R1

Dear Dr. Stein,

We’re pleased to inform you that your manuscript has been judged scientifically suitable for publication and will be formally accepted for publication once it meets all outstanding technical requirements.

Kind regards,

Emily Lund

Academic Editor

PLOS ONE

Additional Editor Comments (optional):

I have reviewed your revisions and believe that you have adequately addressed reviewer comments and strengthened the manuscript.
---

## [Editor Report · Acceptance letter]

26 Jul 2023

PONE-D-23-10564R1 

Episodic future thinking in type 2 diabetes: Further development and validation of the Health Information Thinking control for clinical trials 

Dear Dr. Stein:

I'm pleased to inform you that your manuscript has been deemed suitable for publication in PLOS ONE. Congratulations! Your manuscript is now with our production department. 

Kind regards, 

on behalf of

Dr. Emily Lund 

Academic Editor

PLOS ONE